# Climate gradients, and patterns of biodiversity and biotic homogenization in urban residential yards

**Elizabeth A. Bergey** *, Benjamin E. Whipkey

Oklahoma Biological Survey and Department of Biology, University of Oklahoma, Norman, Oklahoma, United States of America

* lbergey@ou.edu

## Abstract

Residential yards constitute a substantive biodiverse greenspace within urban areas. This biodiversity results from a combination of native and non-native species and can contribute to biotic homogenization. Geographical climatic patterns affect the distribution of native species and may differently affect non-native species. In this study, we examined biodiversity and biotic homogenization patterns of yard-dwelling land snails across 12 towns in Oklahoma and Kansas (USA). The 3 x 4 array of towns incorporated a N-S winter temperature gradient (mean low January temperature range = -8.4 to 0.1˚C) and an E-W annual rainfall gradient (annual rainfall range = 113.8 to 61.3 cm/yr). Ten yards per town were surveyed. We hypothesized that mild winter temperatures and greater annual rainfall would be associated with greater snail abundance and richness, and that the presence of non-native species would contribute to biotic homogenization. Non-native snails were present and often abundant in all towns. Snail communities varied with both rainfall and cold temperature. Contrary to our prediction, snail abundance was inversely related to annual rainfall–likely because drier conditions resulted in greater yard watering that both augmented rainfall and maintained moist conditions. Sørensen similarity between towns for the entire land snail community and for only non-native species both showed distance-decay patterns, with snail composition becoming less similar with increasing distance—patterns resulting from species turnover. The biotic homogenization index also showed a distance-related pattern, such that closer towns were more likely to have biotic homogenization whereas more distant towns tended to have biotic differentiation. These results support the concept that biotic homogenization is more likely regionally and that climatic changes over distance result in species turnover and can reduce spatially broad biotic homogenization.

## Introduction

Biotic homogenization occurs when biotas become more similar through the processes of the loss or expansion of native species and the establishment of non-native species [1] and is considered a global phenomenon [1, 2]. Urbanization has a high potential for producing biotic

**Data Availability Statement:** All relevant data are within the manuscript and its Supporting Information files.

**Funding:** Funding was provided by the University of Oklahoma: SRI funds, Oklahoma Biological

Survey small grants program, and University Libraries (all to EAB). The funders had no role in study design, data collection and analysis, decision to publish, or preparation of the manuscript.

**Competing interests:** The authors have declared that no competing interests exist.

homogenization, given the high levels of disturbance and severe habitat modification, combined with multiple routes of species introduction through the movement of goods and people [3]. Studies of urban biota have supported both biotic homogeneity and biotic differentiation [e.g., 4–7], illustrating trade-offs between changes in native species occurrence and abundance and the establishment of non-native species. Cities may be located in biodiverse regions and in some places retain significant native diversity, including rare species [8, 9], and some native species may benefit from urbanization and thereby increase biotic homogenization [3, 10, 11]. Introduced non-native species adapted to urban environments can establish and increase biotic homogenization [3] or contribute to biotic differentiation [7].

Urbanization produces homogenization of physical habitats, with similar composition of roads, residential areas, commercial areas and aquatic features across cities [3, 12, 13]. Within this overall urban homogenization is a gradient of habitat types from seminatural parks to industrialized areas or city centers with little vegetation. These habitat types act as environmental filters [14] that result in differential patterns of biodiversity [11, 15] and in some cases differential patterns of biotic homogeneity and biotic differentiation [16].

Climatic differences, specifically in temperature regimes and rainfall characteristics, strongly affect species distributions–however the effects of spatial climatic factors on biotic homogenization are poorly known.

To reduce the effects of comparing multiple habitats across cities, this study concentrated on residential yards or gardens that surround dwellings. These yards typically comprise the largest amount of greenspace within urban areas [17, 18] and have high biodiversity of both plants and animals [19–21]. Yards are 'designed' by the residents [21] resulting in management [21–23] and habitat differences among yards as this greenspace serves many resident-based functions, including aesthetic display, recreation, storage of materials, growing food plants and housing animals–or yards may be managed for minimum maintenance.

Land snails are an informative model system to study biotic homogeneity in human-influenced landscapes, including yards. Snail populations in urban areas can be quite speciose, comprised of both local native species and a suite of non-native species [11, 16, 21, 24, 25]. Although snails move slowly on their own, they can be frequent hitch-hikers–dispersing through such routes as the plant trade [24, 26, 27] or landscaping materials.

In this study, we examine the biotic homogeneity of land snails in the residential yards of 12 towns in Oklahoma and Kansas (USA) across two axes of climatic gradient: rainfall and temperature. Both rainfall and cold winter temperatures influence the distribution and abundance of land snails; resulting in a pattern of species replacement among native species [28]. Non-native species often have wide environmental tolerances and this study examines the patterns of native versus non-native species across climatic gradients to determine patterns of biotic homogeneity versus biotic differentiation relative to climatic factors.

We anticipated that the widespread introduction of land snails combined with disturbance-related loss of native species and expansion of tolerant native species would result in significant biotic homogeneity of land snail faunas. We further hypothesized that biotic homogeneity would vary with climatic factors. Specifically, we hypothesized that biotic homogeneity will be lower in towns with different climatic features (more distant towns) than in towns with more similar climates (closer towns).

## Methods

### Selection of towns

Oklahoma has the third highest number of ecoregions of the 50 US states and has the highest number relative to state size (the two states with more ecoregions are California and Texas,

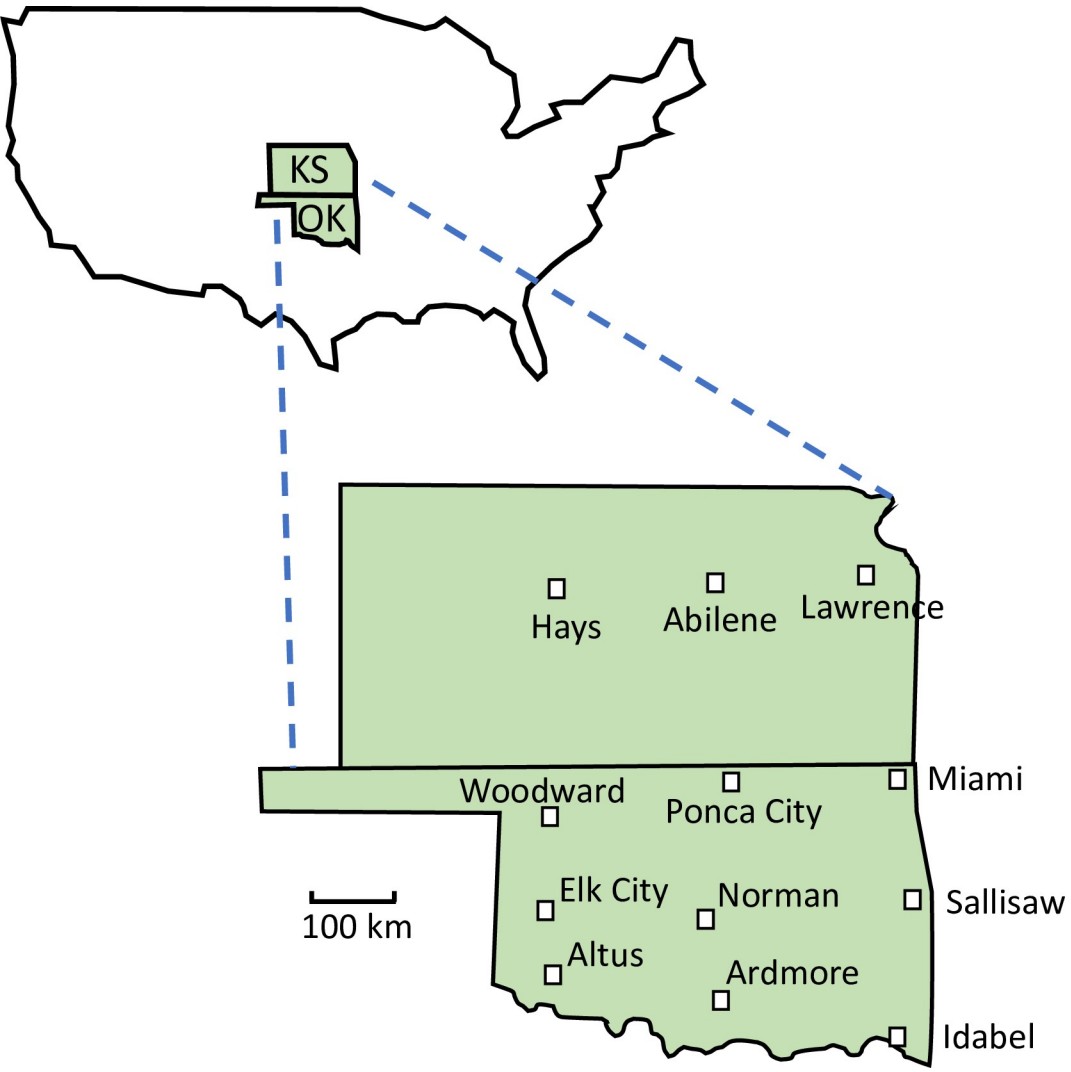

**Fig 1. Map of the 12 surveyed towns in Kansas (KS) and Oklahoma (OK).**

which are much larger than Oklahoma). With the addition of sites in Kansas to increase the temperature range, we captured a relatively large climatic range within a relatively small area.

The twelve selected towns formed a 3 x 4 grid pattern (Fig 1) and encompassed an East-West rainfall gradient (Fig 2A and S1 Table) and a North-South temperature gradient (Fig 2B). Town populations ranged from about 6,500 to 122,000 (S1 Table). Climate data for towns were characterized by the mean annual rainfall and the mean monthly low temperature of the coldest month (January) over a 20-year period (1999–2018). Climate data were obtained using the NOWData database [NOAA on-line Weather Data; 29], where data were available for all towns except Ardmore, for which data from the Oklahoma Mesonet [30] were used.

## Selection of survey yards

We relied on three methods to find yards to survey: previous contacts, contacting the local branch of state agricultural extension program prior to our visit and asking them post a Facebook message eliciting participants, and snowballing (asking contacts or participants for

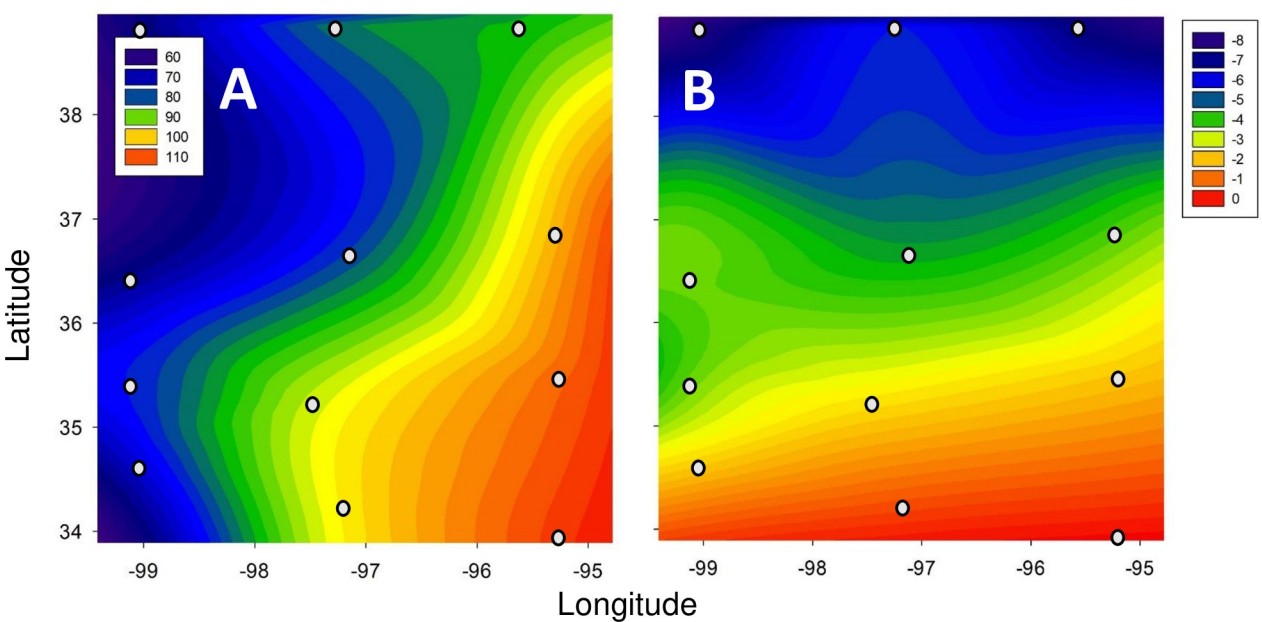

**Fig 2. Climatic gradients across the study area.** (**A**) Total average yearly rainfall (in cm/yr) increases from east to west and (**B**) winter temperatures (˚C: shown as mean monthly low temperature, which occurred in Jan) decreases from south to north. Small circles show locations of surveyed towns.

further recommendations). We talked to residents, describing our survey rationale and methods and asked about yard management. Residents were provided with a one-page fact sheet with information about land snail ecology and urban land snails. Following surveys, we again met with residents to go over the diversity of snails found in their yards (a surprise to many residents). This research project was determined to not be human research by the University of Oklahoma IRB and, consequently, the IRB also waived the need for informed consent.

Surveys were conducted between in the spring and early summer (inclusive dates were 28 April to 29 June) during 2017 and 2018 (S1 Table) and consisted of one, 80-minute timed visual survey of each yard. Habitats surveyed included under leaf litter, especially leaf litter under shrubs; under wooden boards, pots on the ground, and debris; along raised edges (e.g., along walls of structures, plant bed edging, and driveways); and beneath ground-covering plants. In addition to visual searching, a surface soil sample, measuring approximately 0.5 l, was collected in at least two areas in each yard with visible accumulations of micro-snail shells or, where such accumulations were not apparent, from habitats where micro-snails or their shells were commonly found (e.g., friable soil at the base of buildings). Visual surveys and soil collections included both front and back yards.

Larger live snails were generally provisionally identified, counted and released; empty shells and unidentified live snails and were retained for definitive identification, and for matching with the provisionally identified live snails. Examples of slugs and most live micro-snails were also collected. Collected soil surface samples were dried and checked for micro-snails under magnification. Occasionally, snails were reared to enable identification (e.g., when only immature polygyrids were found).

The following snail identification guides were consulted: Pilsbry [31]; Burch [32]; Hubricht [28 and 33]; Perez and Cordeiro [34]; Nekola and Coles [35]; Grimm, Forsyth [36]; and Hotopp et al. [37]. Difficulties arose in the identification for a few taxa and are summarized as follows: *Strobilops* spp. included both *S. texasiana* and *S. labyrinthica*, *Gastrocopta armifera* and *Gastrocopta abbreviata* were combined, as were *Vallonia pulchella* and *V. excentrica* (both

groups had many intermediate forms), succinids were identified to family, and individuals that combined characteristics of both *Gastrocopta pentodon* and *Gastrocopta tappanianna* were designated as *G. pentodon*. Immature snails and shells were counted as unidentified.

The paucity of historic records in the regions of the 12 towns combined with the wide climatic range across towns made determination of native versus non-native species challenging. A conservative approach was used; specifically, species recorded as present in Oklahoma and/or Kansas in Hubricht [28] were considered native, even though many species have restricted distributions with the state. For example, *Anguispira alternata* was considered native in both states even though its native distribution is restricted to the eastern-most tier of towns [28] and its presence in the middle of Oklahoma [Norman: 23] is a disjunct, likely introduced population.

Distance between pairs of towns was measured between town centers using Google Earth Pro.

## Data analysis

Data from visual surveys and soil samples were combined; hence abundance and richness include both live snails and shells, unless otherwise specified. In addition, data from the 10 surveyed yards in each town were pooled to produce a single snail composition data set per town.

Patterns between climatic factors (mean monthly low temperature for January and annual rainfall) and snail abundance and richness were analyzed using linear regression (Systat within SigmaPlot 12.0 software). Effects of town population size on snail richness were also tested using linear regression. Taxonomic composition of snail assemblages across towns was compared using one-way PERMANOVA (Primer 6 plus Permanova+; Primer-E software). Snail compositions within towns were characterized using SIMPER (Primer 6) to determine the species contributing 90% of the within-town Bray Curtis similarity. The relationships between snail assemblages and climatic variables (mean cold temperature and annual rainfall) were analyzed using DistLM (Distance-based linear models; Primer 6 plus Permanova+).

Snail assemblages among the 66 combinations of pairs of towns were compared using three similarity indices: Sørensen and Simpson indices are both presence/absence indices and Bray-Curtis is an abundance-based metric. Equations and references for all three similarity indices are found in Koleff et al. [38]. Sørensen and Bray-Curtis indices were calculated using Primer 6 and the Simpson index was manually calculated using an Excel spreadsheet and the equation from Koleff et al. [38].

The Sørensen index was used to assess changes in assemblage similarity across distance (= distance decay), which was analyzed using exponential decay regression (Systat software). Although the Jaccard Index is often used for distance decay, the Sørensen and Jaccard indices are similar and a regression of our data using the 66 possible pairwise comparisons produced an $R^2$ of 0.991. Sørensen similarity is also commonly used similarity index for measuring beta diversity, indicating biotic homogenization and, in combination with the Simpson similarity index, is used to distinguish spatial turnover and spatial nestedness components of biotic homogenization [39, 40].

The effect of non-native species on the homogenization versus differentiation of communities was examined using the homogenization index described in Horsák, Lososová (16). This index is the difference between the similarity based on all taxa present and that based on only native species, and produces values ranging from -1 to +1. Positive values indicate that non-native species contribute to homogenization by having a relatively low similarity whereas negative values indicate differentiation, with the similarity within non-native species being higher than for all species combined [16].

## Results

### Characterization of snail faunas

Snails were found in all 120 yards included in the survey. The total survey count was 18,079 live snails and shells that comprised 49 taxa (S2 Table). Twenty taxa were not native, comprising 9 extralimital species (native elsewhere in the country) and 11 alien species (not native to the United States).

Snail abundance, as the combined count of shells and live snails in the 10 yards per town, averaged 1508 snails (SE = 216.9) per town, or 150.8 snails/yard. Towns varied in snail abundance (Fig 3A) and ranged between catches of 568 snails in Abilene and 3290 snails in Altus.

Abundance by collection type (S1 Fig) averaged 46.8% (SE = 3.53%) for visually collected shells, 27.6% (SE = 4.8%) for counted and released live snails and 25.6% (SE = 4.4%) for shells in the soil sample (note: both the shells and soil sample included some live microsnails that were not included in the live snail count). Live snails were relatively more abundant in the eastern tier of towns and microsnails in the soil samples were relatively more abundant in the western tier of towns (S1 Fig).

Like snail abundance, taxonomic richness varied among towns (Fig 3B). Richness averaged 21.7 taxa (SE = 1.7) per town. Snail taxonomic richness ranged from a low of 13 taxa in Hays to a high of 26 taxa in Ardmore. Taxonomic richness was not correlated with town population size (regression: $F_{1,10}$ = 0.376; p = 0.553; $R^2$ = 0.04).

Non-native species were present in all 12 towns and in 111 of 120 yards (versus native species, which were found in all towns and 119 yards). Non-native snail abundance (Fig 3A) averaged 504.5 snails per town (SE = 72.6) and ranged from 137 snails in Elk City to 922 snails in Norman. Percent abundance of non-native to all identified snails ranged from 8.9% in Elk City to 66.3% in Sallisaw. Two other towns had non-native snails comprising over 50% abundance: Hays (57.6%) and Norman (51.7%). Taxonomic richness of non-native snails across towns averaged 5.8 (SE = 0.7) species (Fig 3B) and ranged from one taxon, *Vallonia* spp. (*V. pulchella* and *V. excentrica*, which were both present) in Hays to 9 species in both Altus and Ardmore.

### Spatial patterns of snail abundance and taxonomic richness

Snail abundance was relatively low across towns in the northern tier (Hays, Abilene, Lawrence), the eastern tier (Lawrence, Miami, Sallisaw, Idabel) and the south-central town of Ardmore, and was highest in southwestern Oklahoma (Fig 4A). The distribution of live snail abundance (snails counted and released during surveys) was similar to the pattern of all snails and shells (compare Fig 4A and S2 Fig), with relatively low abundance across the northern and eastern tiers, except for the higher relative abundance of live snails in Lawrence (the NE-most town). Centrally located Norman also had a relatively high abundance of live snails compared to all snails. Both live snails and all snails had the highest abundance in the SW-most town of Altus.

The regression of annual rainfall and low winter temperature with snail abundance was significant ($F_{2,9}$ = 7.253; p = 0.013). Snail abundance was negatively associated with increasing rainfall (Fig 5A; $R^2$ = 0.26) and positively but weakly associated with higher temperatures during the winter (Fig 5B; $R^2$ = 0.03).

Snail species richness displayed a spatially complex pattern (Fig 4B). Richness was low in the northwest and southeast portions of the survey and high in the southwest, which is similar to the pattern of total snail abundance (Fig 4B versus 4A). Snail richness showed little association with either rainfall or low winter temperatures (Fig 5C and 5D; $F_{2,9}$ = 0.515; p = 0.62).

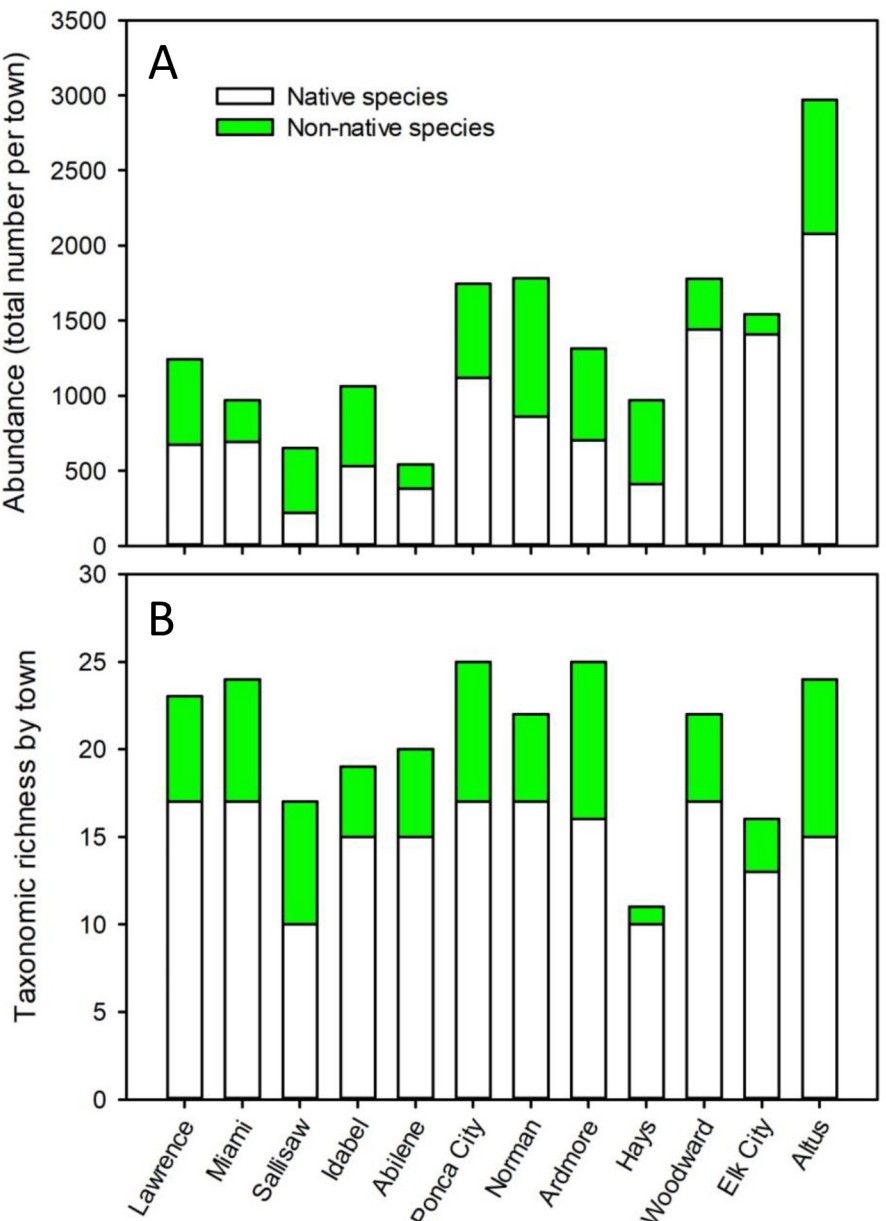

**Fig 3. Native (white) and non-native (green) composition of snail assemblages among towns.** Towns are arranged from north to south, starting with the east tier, then the mid- and west tiers. **A**: Total snail + (shell) abundance. **B**: Taxonomic richness.

## Spatial patterns of taxonomic composition

Snail composition based on species' abundance differed among towns (one-way PERMA-NOVA: Pseudo-$F_{11,108}$ = 5.096; p ≤ 0.001). Most pairwise comparisons of snail species' abundance were significantly different between town pairs, with 49 of 66 comparisons having p ≤ 0.001 and t values ranging between 1.839 and 3.405 (S3 Table and Fig 6). Eight comparisons had p values of 0.002 to 0.007 (t = 1.592–2.228). The remaining 9 paired comparisons had p values ranging from 0.015 to 0.65. Towns without significantly different assemblages (i.e., p > 0.05) were located in central to southern areas of the study (the six pairwise combinations

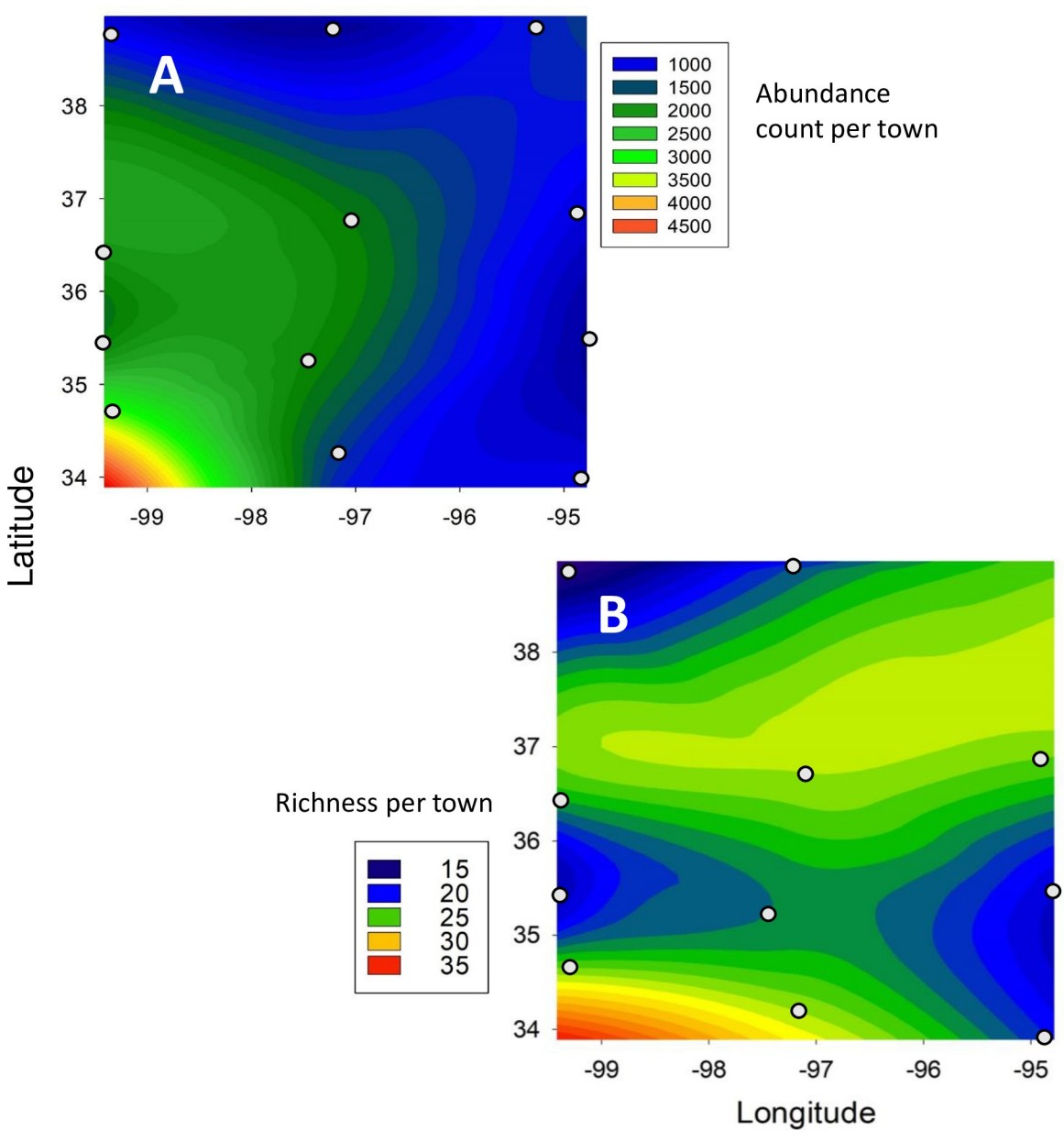

**Fig 4. Patterns of snail assemblage attributes across the survey area.** (**A**) Snail and shell abundance. (**B**) Species richness. Small circles show location of towns included in the survey.

among Ponca City, Norman, Ardmore, and Idabel) or were western (the combination of Woodward and Elk City; Fig 6).

A distance decay pattern was evident as the Sørensen similarity of snail assemblages decreased with increasing distance among towns (exponential decay regression: p < 0.001, $R^2$ = 0.35; Fig 7). Following Baselga [39], this distance decay relationship is also illustrated as dissimilarity, which increases with distance (linear regression: Fig 8A). Simpsons dissimilarity, a measure of species replacement, likewise increased with distance (p < 0.001, $R^2$ = 0.389; Fig 8B). Nestedness (within the dissimilarity index) was little related to distance among towns (p = 0.50, $R^2$ = 0.007; Fig 8C).

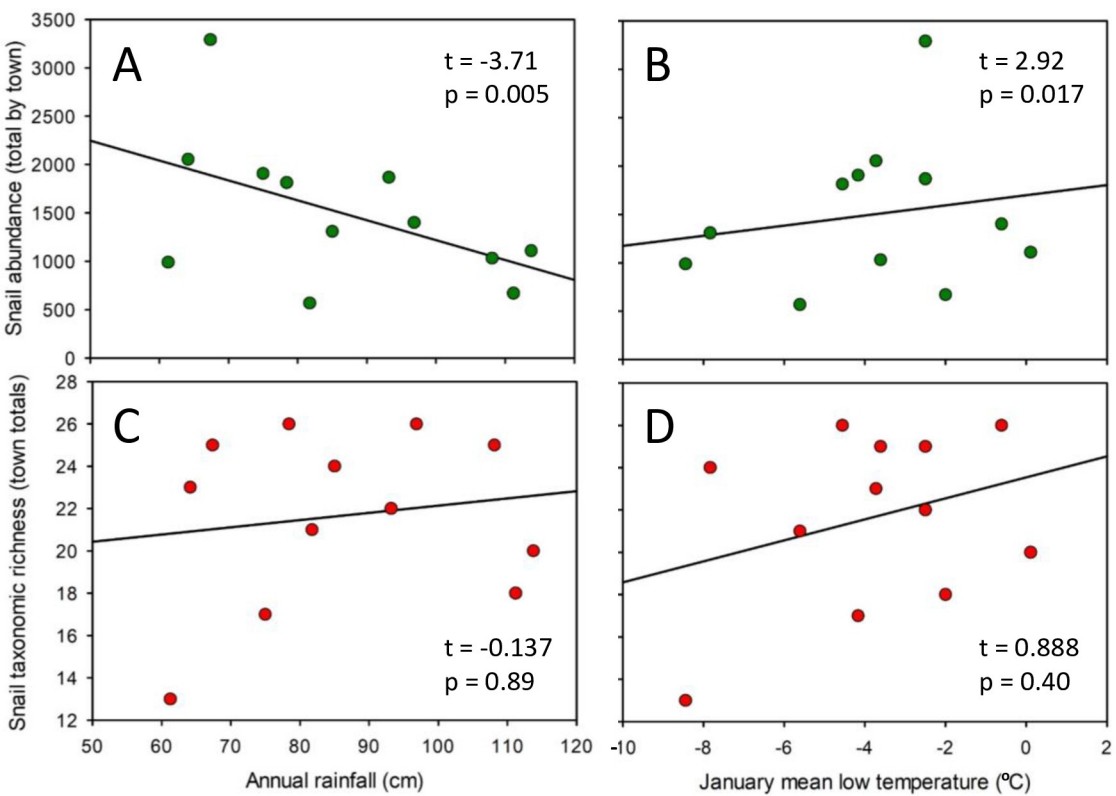

**Fig 5. Relationships between abiotic factors (annual total rainfall and monthly mean low temperatures in January) and snail abundance and richness across the 12 surveyed towns.** Snail abundance (green circles): **A** is rainfall; **B** shows low temperatures. Snail species richness (red circles): **C** is rainfall; **D** shows low temperatures. Multiple linear regression results are displayed on each graph.

Variation in snail assemblage composition among towns was explained by both annual rainfall (DistLM: $R^2 = 0.257$; Table 1) and coldest mean monthly temperature ($R^2 = 0.379$). Together, rainfall and cold temperature explained 50.1% of the variation in taxonomic composition among towns.

Snail species typical of yards in each town (based on SIMPER analysis; Fig 6) range from species that are abundant in all towns (i.e., *Hawaiia minuscula* and *Zonitoides arboreus*) to species that are typical of only a single town (i.e., *Mesodon thyroides*, *Xylotrema fosteri*, *Polygyra cereolus*, *Gastrocopta procera*, and *Gastrocopta pellucida*). Patterns of typical species vary in the N-S, E-W gradients. For example, *Vallonia* spp. are typical of only the northern (Kansas) tier of towns, *Gastrocopta cristata* was a typical species in only the western towns, and *Ventridens demissus* was abundant in the mid to southeastern sites. Excluding *Hawaiia minuscula* and *Zonitoides arboreus* (typical of all 12 towns), distributions of typical species were greater than indicated in Fig 6 because low abundance and/or spotty distributions within towns result in low contributions to Bray-Curtis similarity within towns; that is not all species present were designated as typical species based on SIMPER analysis.

## Non-native snails and biotic homogeneity

Non-native snail species were among the typical species in 11 of 12 towns (Fig 6). The Sørensen dissimilarity of non-native snail assemblages increased with increasing distance among towns (p = 0.047, $R^2 = 0.06$; Fig 8D). Simpsons dissimilarity, a measure of species replacement,

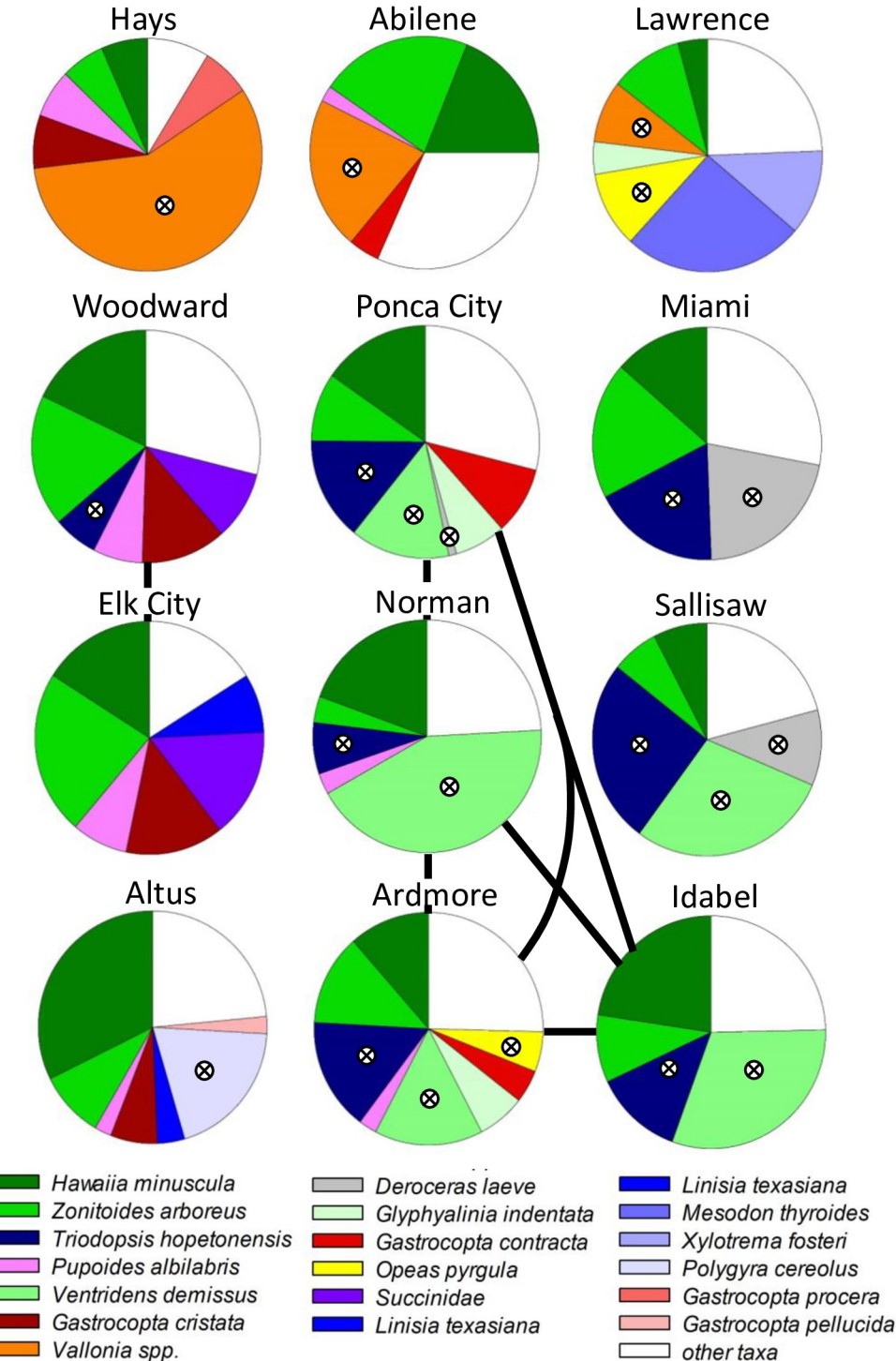

**Fig 6. Relative abundance of typical snail species comprising 90% of the Bray-Curtis similarity within each town, based on SIMPER analysis.** Grid pattern of pie diagrams reflects the grid pattern of towns. Snail families are color-coded: greens = Zonitidae; blues = Polygyridae; reds = Vertiginidae; all other families (other colors) are monospecific within the typical taxa for each town. White areas are the pooled relative abundance of all other snail species. Crossed circles indicate non-native species. Black lines connecting pie diagrams indicate non-significant differences in snail composition between town pairs.

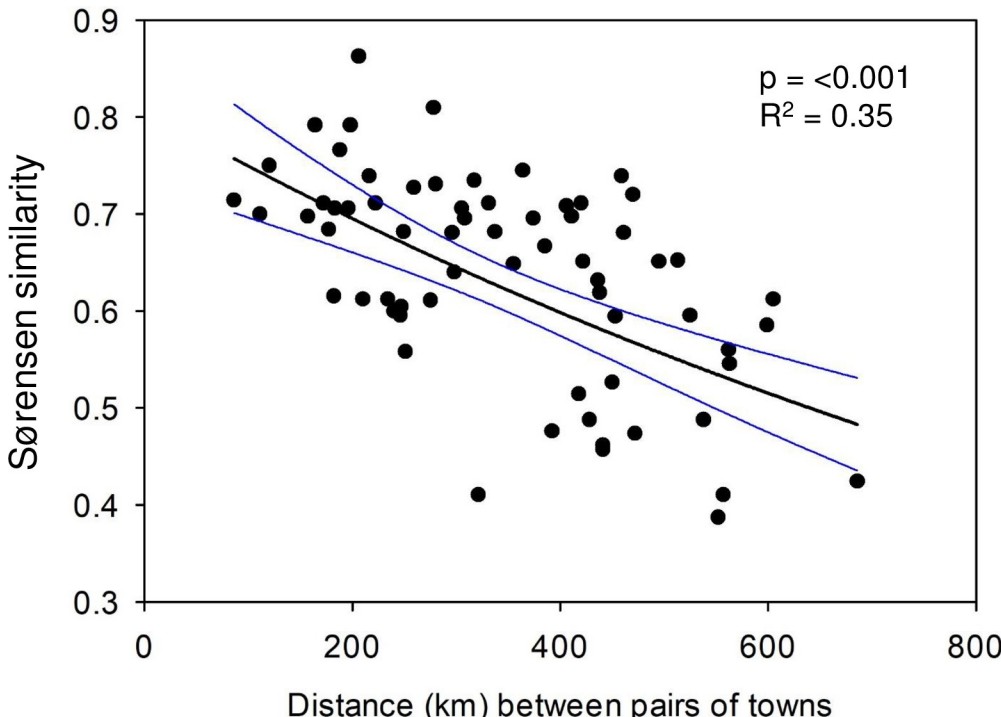

**Fig 7. Distance decay relationship, as illustrated by the exponential decay regression of the distance between pairs towns and the corresponding Sørensen similarity of the snail faunas (N = 66).** The black line is the regression line and the flanking blue lines are the 95% confidence intervals.

also increased with distance (p = 0.004, $R^2$ = 0.123; Fig 8E). Nestedness (within the dissimilarity index) was little related to distance among towns (p = 030, $R^2$ = 0.017; Fig 8F).

The homogenization index among pairs of towns ranged from moderate homogeneity (0.231 in a scale of -1 to +1) to high differentiation (-0.516). Of the 66 pairs of towns, 26 pairs supported homogenization, 39 supported differentiation, and 1 was neutral (S3 Table). The regression between the distance between towns and the homogenization index indicated a distance-related relationship, in which closer towns were more likely to show biotic homogenization and more distant towns were more likely to have biotic differentiation (p = 0.004; $R^2$ = 0.11; Fig 9).

## Discussion

### Distance-decay

Land snail populations in residential yards were speciose and displayed an overall distance-decay spatial pattern based on similarity of taxonomic composition; that is, reduced similarity with increasing distance among towns. Distance decay patterns result from a combination of spatial variation in environmental factors, and the dispersal ability and environmental tolerances of organisms [41]. Distance decay patterns have been observed in a wide variety of organisms [42]; including trees [41], predatory invertebrates [43], and aquatic communities [44, 45].

The most apparent environmental factor that could cause a distance decay pattern in our study system is climatic variation among towns–specifically gradients of westward diminishing rainfall and northward colder winter temperatures. Both rainfall [46–49] and cold

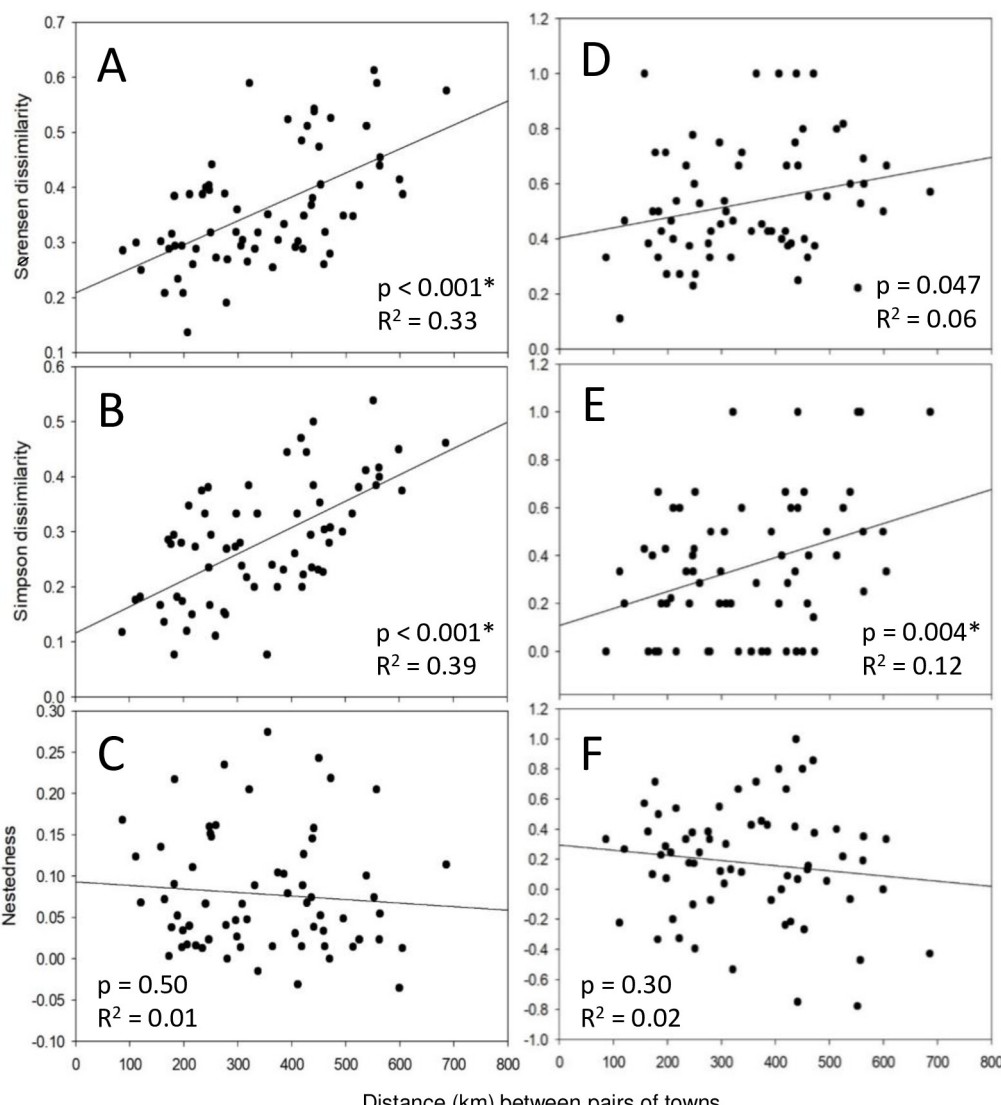

**Fig 8. Relationships between dissimilarity indices and distance between pairs of towns (N = 66). A-C**: All taxa identified to species. **D-F**: non-native taxa only. The Sørensen dissimilarity index (**A, D**), which is a presence-absence metric, is partitioned to Simpson's dissimilarity index (a measure of species replacement; **B, E**) and nestedness (**C, F**). Regression p and $R^2$ values are given on each graph. A Bonferroni-corrected α = 0.025 was used to adjust for using two sets of related data (total and non-native). Significant p-values are denoted by '*'.

**Table 1. Statistical summary of distance-based linear model between climatic variables and land snail community composition.**

| Marginal tests | | | |
|---|---|---|---|
| **Variable** | **Pseudo-F** | **P** | **Proportion explained ($R^2$)** |
| Cold temperature | 6.095 | 0.0002 | 0.379 |
| Annual rainfall | 3.4544 | 0.0175 | 0.257 |
| **Sequential test** | | | |
| Temperature then rainfall | | | 0.501 |

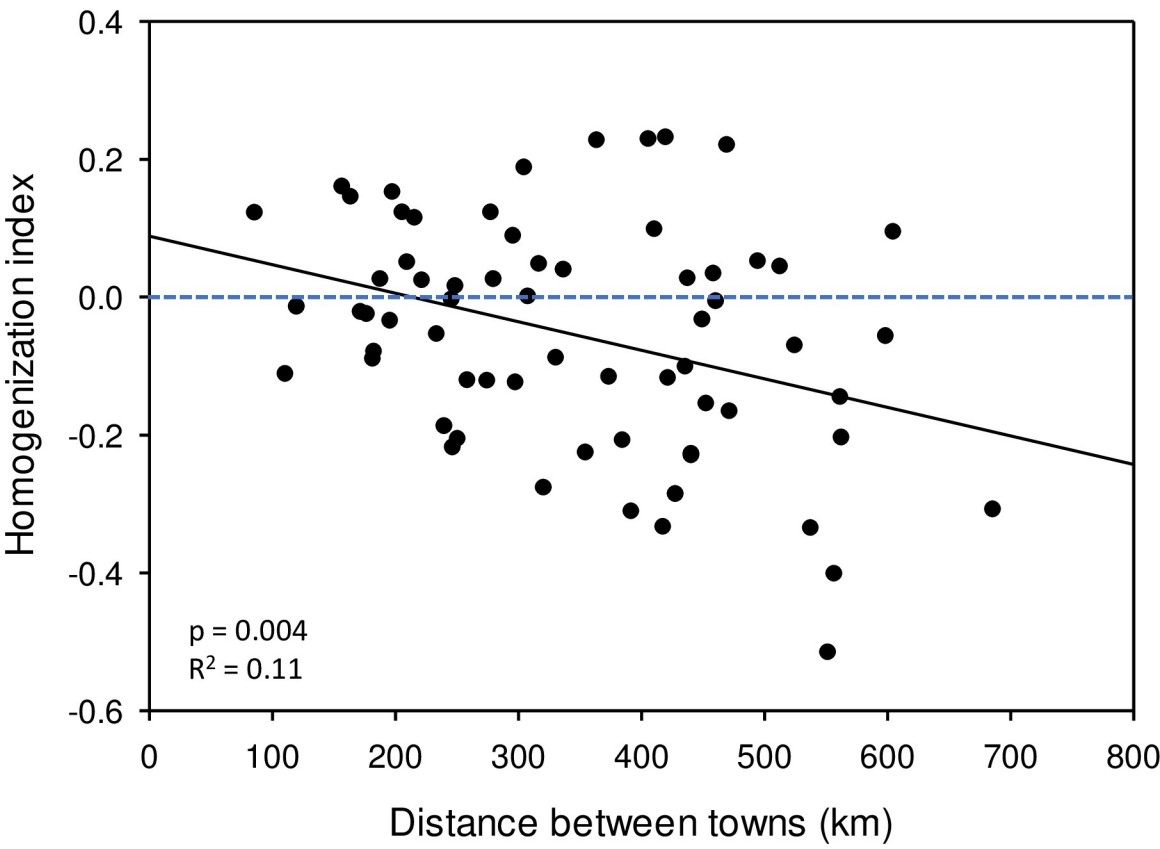

**Fig 9. Relationships between the homogenization index and distance between pairs of towns (N = 66).** The dashed blue horizontal line is the transition between biotic homogenization ($> 0$) and biotic differentiation ($< 0$).

temperatures [50, 51] can affect land snail composition, and indeed both climate variables explained much of the variation in taxonomic composition among towns, together explaining one-half (50.11%) of this variation.

Native snails should be adapted to local seasonal and year-to-year fluctuations in temperature, whereas some non-native snails may be strongly affected by ambient climatic conditions because non-natives may originate from regions with a different climate and, as a consequence, be poorly adapted to local conditions—although many non-native snails are tolerant generalists. An incongruence between climatic tolerance and species' presence may occur in species that are dispersed primarily by humans, as unintentional human-assisted dispersal can be both rapid and long distance. Indeed, climate has become a driver in worldwide land snail distribution patterns as a result of increased human-mediated dispersal [52].

Cold temperatures decrease the daily and seasonal activity periods of snails and very cold temperatures can be lethal. Although thermal limits are poorly known for most snail species, the thermal tolerance of one non-native from this study, namely *Cornu aspersum*, has been studied in detail [e.g., 53]. *Cornu* has limited tolerance to freezing temperatures, and indeed, we found *Cornu* in only the southern tier of sites, specifically in Altus, where the mean low temperature in January is -2.5°C. Populations of *Cornu* also live in Norman [23], which has a similar low mean temperature, and in several urban areas in New Mexico [54], where winter low temperatures are colder. Although physiologically ill-adapted to these lower temperatures, the species and other freeze-susceptible snails likely persist through behavioral adaptation,

especially in finding protective microhabitats. Such habitats include areas with herbaceous cover, leaf litter and soil textures amenable for burrowing [55, 56], characteristics that may also benefit native species in less human-modified habitats [16]. Urban areas may provide additional 'heated' habitats, including areas adjacent to thermally leaky buildings [e.g., 23] and decomposing compost piles [56]. Exposure can also be avoided by living on potted plants that are brought indoors during the winter. Additionally, indoor gardens or greenhouses can host a variety of snail species [27, 57, 58] and be sources of distribution into the local environment.

In contrast to our expectations, snail abundance decreased with increasing rain. In residential yards, the 'rainfall' experienced in the habitat is a combination of actual rainfall, infiltrating runoff from impervious surfaces, and applied rainfall; that is, the watering of vegetation by residents. Yard watering not only augments rainfall but provides a predictably moist environment over the often-dry summer period in the Great Plains region. The effect of watering was especially evident in the south-most tier of sites, in which the drier SW site (Altus; mean annual rainfall = 67 cm) had higher species richness and abundance than the more mesic SE site (Idabel; mean annual rainfall = 114 cm). In the drier Altus, one-half of the surveyed yards were on a regular watering scheme of 2–3 times per week, whereas in Idabel, only one-fourth of the yards received this frequency of added water. Additionally, the more mesic Idabel site has a high water table and few residents had watered their yards during the still-mild summer of the town's survey (27–29 June), resulting in green plants and a very dry soil surface–conditions that contributed to the inverse relationship between rainfall and snail richness and abundance.

The dispersal ability of organisms can also affect the distance decay relationship, in that restricted dispersal results in species replacement over distance–strengthening the distance-decay relationship. Urban snail assemblages as a group, have complex dispersal patterns that combine slow self-dispersal by crawling [23] with long-distance jumps, as snails are incidentally transported by the movement of goods and products.

## Biotic homogeneity

All 12 towns had significant non-native snail populations, often comprising several species, supporting the possibility of biotic homogeneity. Habitat destruction and continuing disturbance are characteristic of yards and although it's probable that native species loss occurred, such loss could not be documented, as pre-urban and local faunas, in general, are poorly known in the central Great Plains of North America.

Distance-decay relationships are a useful tool to indicate spatial patterns in biotic homogeneity. If biotic homogeneity is high, spatial similarity of biotas would be reflected in a weak distance decay relationship. The strong distance-decay relationship across the entire snail fauna indicates little overall biotic homogeneity in yard-dwelling snails. Examination of the distance-decay relationship for only non-native species demonstrates an underlaying pattern more supportive of biotic homogeneity (e.g., a much less robust distance-decay relationship relative to the all-taxa dataset). Species replacement is apparent for both entire snail assemblages and for the non-native subset but is much stronger for the entire assemblage. Species replacement is consistent with spatial patterns of climate and (water) management and mirrors spatial changes in the native fauna.

Biotic homogenization and biotic differentiation were related to the distance between towns. In general, biotic homogenization was more frequently associated with nearby towns, whereas distant towns had higher frequency of biotic differentiation. This spatial relationship is consistent with the spatial patterns of climate and species turnover. That is, as distance between towns increased, climate differences increased–resulting in snail species turnover and increased biotic differentiation.

## Supporting information

**S1 Fig. Percent contribution of each sample type to the total abundance in each of the 12 surveyed towns.** Towns are shown in their relative N-S and E-W orientations. 'Shells' included both macro- and micro- snails—and includes some live microsnails, 'Live snails' were field-identified snails and slugs that were released on site or reared for identification, "Soil sample' included primarily microsnails–mostly shells, but also live snails.
(PDF)

**S2 Fig. Spatial pattern of live snail abundance across the 12 surveyed towns.** The graph is based on snails counted and released during the visual survey and undercounts live microsnails that were included in the shell or soil samples.
(PDF)

**S1 Table. Twelve towns included in the regional survey.** Population is based on the US Census Bureau (2018) estimates for 2016. Climatic data are means over a 20-yr period. Latitude and longitude are mean values for the ten yards sampled in each town.
(PDF)

**S2 Table. Taxa list, native/non-native designation relative to Oklahoma and Kansas, and total count of each taxon for the 10 yards in the 12 towns in the survey.**
(PDF)

**S3 Table. Pairwise comparison results following PERMANOVA test of the Bray Curtis similarity of the yard-dwelling land snail populations in 12 towns, which produced 66 pair-wise comparisons.**
(PDF)

## Acknowledgments

We are extremely grateful to the many residents who allowed us to survey their yards and appreciated their frequent offers of hospitality. We also thank the state extension agents and others who helped find participating residents. Matthew Carman helped with field work, snail sorting and identification.

## Author Contributions

**Conceptualization:** Elizabeth A. Bergey.

**Data curation:** Elizabeth A. Bergey.

**Formal analysis:** Elizabeth A. Bergey.

**Investigation:** Elizabeth A. Bergey, Benjamin E. Whipkey.

**Methodology:** Elizabeth A. Bergey.

**Writing – original draft:** Elizabeth A. Bergey.

**Writing – review & editing:** Benjamin E. Whipkey.

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
