## [Decision Letter · Decision Letter 0]

23 Jun 2020

PONE-D-20-15300

Climate gradients, and patterns of biodiversity and biotic homogenization in urban residential yards

PLOS ONE

Dear Dr. Bergey,

Thank you for submitting your manuscript to PLOS ONE. After careful consideration, we feel that it has merit but does not fully meet PLOS ONE’s publication criteria as it currently stands. Therefore, we invite you to submit a revised version of the manuscript that addresses the points raised during the review process.

Your manuscript has been reviewed by two reviewers. As you can see, both reviewers think that the study was well designed and executed. However, there are important points raised by Reviewer 1 that may require you to provide more clarification on the sampling methodology and data analysis. Hence, please respond to all the constructive comments by both reviewers, in particular with a better explanation of the methodology and explore the suggested additional analyses.

We look forward to receiving your revised manuscript.

Kind regards,

Thor-Seng Liew, Ph.D.

Academic Editor

PLOS ONE

Additional Editor Comments:

Your manuscript has been reviewed by two reviewers. As you can see, both reviewers think that the study was well designed and executed. However, there are important points raised by Reviewer 1 that may require you to provide more clarification on the sampling methodology and data analysis. Hence, please respond to all the constructive comments by both reviewers, in particular with a better explanation of the methodology and explore the suggested additional analyses.

2. Please provide additional details regarding participant consent.

In the ethics statement in the Methods and online submission information, please ensure that you have specified (i) whether consent was informed and (ii) what type you obtained (for instance, written or verbal, and if verbal, how it was documented and witnessed).

If the need for consent was waived by the ethics committee, please include this information.”

3. In your Methods section, please state where the participants were recruited for your study.

4. In your Methods section, please provide additional location information of the study sites, including geographic coordinates for the data set if available.

5. We note that Figure 1 in your submission contains a map image which may be copyrighted.

We require you to either (a) present written permission from the copyright holder to publish these figures specifically under the CC BY 4.0 license, or (b) remove the figure from your submission:

b. If you are unable to obtain permission from the original copyright holder to publish this figure under the CC BY 4.0 license or if the copyright holder’s requirements are incompatible with the CC BY 4.0 license, please either i) remove the figure or ii) supply a replacement figure that complies with the CC BY 4.0 license. Please check copyright information on all replacement figures and update the figure caption with source information. If applicable, please specify in the figure caption text when a figure is similar but not identical to the original image and is therefore for illustrative purposes only.

Reviewers' comments:

Reviewer's Responses to Questions

**Comments to the Author**

1. Is the manuscript technically sound, and do the data support the conclusions?

Reviewer #1: Partly

Reviewer #2: Yes

2. Has the statistical analysis been performed appropriately and rigorously? 

Reviewer #1: N/A

Reviewer #2: Yes

3. Have the authors made all data underlying the findings in their manuscript fully available?

Reviewer #1: Yes

Reviewer #2: Yes

4. Is the manuscript presented in an intelligible fashion and written in standard English?

Reviewer #1: Yes

Reviewer #2: Yes

5. Review Comments to the Author

Reviewer #1: Dear authors,

I like your study, its well-designed, clearly aimed, the analyses are adequate and the conclusions are well supported. The writing is concise and well structured.

I have, however, several comments and suggestions how to improve the quality of your work. There is nothing severe, but some points should be considered -- there is certainly room for improvements and corrections.

1) Description of sampling methods.

I miss some details about the sampling. Specifically, if you collected leaf litter in all sampled yards to capture minute snails. If so, how many litters per yards, it should be standardized in terms of sample size, stratification and the total area samples. If this not fully comparable among yards, it should be tested to convince readers that there is no explanatory power of these methodological differences. I also miss year(s), and dates of sampling. Were the yards sampled in the same year/season? The thing here is to prevent any sampling scheme structured along the climate gradients – e.g. all towns in wetter areas are sampled in one year etc. I do not think this was a case, but it is important to make this point clear for readers. How many times was the given yard sampled (once?).

Only until looking at the figure 5, I was not sure if you analysed individual yards or polled them per town. Please, be specific on this in the Methods. I am not sure whether we are not losing too much information by polling yards and using town as observations to be analysed. I understand that this is a simple way how to avoid pseudoreplications, but there are (not so complicated) statistical methods how to solve this. Maybe you should consider to analysed the data at yard scale, which definitely would increase statistical power, but also enable to test some yard specific characteristics as for example yard management (you asked people about it) watering, size etc.

2) Live snails vs empty shells.

It is not clear if only live snails were used to the analyses or snail individuals, including empty shells, were polled and used in all analyses. If so, some test that the live/empty ration is not changes along climate would be useful. Empty shell decomposition rate is highly variable based on climate.

3) The effect of town size.

I realized that the study towns exhibited quite a broad range of sizes, assuming based on town populations ranging from 6.5 to 122K. This is very likely to have an effect on species richness (just as simple SAR pattern) but also more people means more transport, resulting in higher probability of non-native species introduction. I would like to encourage you to test/explore this in your data and to show whether this is or is not an issue.

There might be two levels – the effect of town size/population and the total area cover by the sampled yards within a town. Maybe an interesting analysis, additional to total number of species per town, could be to correlated compositional dissimilarity among yards of the same town with these two size measures and also climate. Are yards in smaller towns more similar or is there any effect of climate (e.g. less humid towns are more homogeneous)?

Related to this point, you should use more than a single regression method as some explanatory variables are very likely to be intercorrelated. Using a simple multilinear regression model we can test the effect of climate and size (population density) on numbers of species and snail abundance. If the size of a town is added as the first variable, then climate is tested on a residual variation, independent of this possibly confounding factor.

4) Distance-decay and Figure 7.

Please improve your theoretical background about distance-decay relationship. The theory says that it is either power law or exponential decay, depending on spatial scale and the homogeneity of species pool. A linear relationship is against the theory and needs to be explained. A good paper to read on this: Nekola, J. C., & McGill, B. J. (2014). Scale dependency in the functional form of the distance decay relationship. Ecography, 37(4), 309-320.

Looking at the data, it is quite obvious that in two cases (A, B), showing a significant decrease of similarity with distance, either power law or exponential function will have a better fit than a linear model.

Please change y-axes to be similarity, so we can see the decrease with distance. You also need to use Bonferroni/Holm correction of the significance cut level for multiple testing as there are 6 tests run against distance. The correction will certainly leave out only a marginally significant relationship at the D, having visually no decay of similarity with distance. When insignificant, avoid showing a fitting curve, it does not make sense.

I am hoping you will find these comments useful and wish you good luck with your research.

Reviewer #2: I really enjoyed this paper and I'm sure it fills a void in the literature, for several reasons. One is that land snails (and slugs) and grossly understudied. This is esp true for urban habitats. Urban ecology and urban conservation is a rapidly growing field and we know very little about land snails that inhabit cities. In terms of your study, your methodology and research design are excellent. Again, we know very little about climate and precipitation gradients in land snails and this study is the first to tease apart some important patterns. I was esp intrigued by the role that watering lawns seems to play. The distance-decay and homogenization patterns are less surprising but still need to be documented. Your analysis is also very good. I like that you include abundance (Bray-Curtis) and not just species similarity indices. You also did a good job with the taxonomic ID, often a difficult task with snails. Few people rear juveniles to confirm ID.

The paper is well researched, well written and you know the literature well. I just have a couple of minor suggestions. For line 119, I am not sure what you mean by "composite" soil sample. I am guessing that you sampled in various places but please expand. Secondly, in my copy the text on some figs (1,2,4,6) is blurred. Perhaps that is just my copy but you might check.

6. PLOS authors have the option to publish the peer review history of their article (what does this mean?). If published, this will include your full peer review and any attached files.

Reviewer #1: No

Reviewer #2: Yes: Michael L. McKinney

---

## [Author Response · Author response to Decision Letter 0]

21 Jul 2020

Reply to Reviewers

Reviewer #1: Dear authors,

I like your study, its well-designed, clearly aimed, the analyses are adequate and the conclusions are well supported. The writing is concise and well structured.

I have, however, several comments and suggestions how to improve the quality of your work. There is nothing severe, but some points should be considered -- there is certainly room for improvements and corrections.

1) Description of sampling methods.

I miss some details about the sampling. Specifically, if you collected leaf litter in all sampled yards to capture minute snails. If so, how many litters per yards, it should be standardized in terms of sample size, stratification and the total area samples. If this not fully comparable among yards, it should be tested to convince readers that there is no explanatory power of these methodological differences.

--- I added the amount (‘approximately 0.5 l’) and the stratified nature of sampling a sample (‘All visual surveys and soil surface collections included both front and back yards’). I also added the relative distribution of snail abundance among collection types (visual counts of shells and of live snails – and snails/shells in the soil samples) to the Results and added a supplemental figure showing catch by collection type for each town; Suppl. Figure 1).

I also miss year(s), and dates of sampling. Were the yards sampled in the same year/season? The thing here is to prevent any sampling scheme structured along the climate gradients – e.g. all towns in wetter areas are sampled in one year etc. I do not think this was a case, but it is important to make this point clear for readers. How many times was the given yard sampled (once?).

--- The general sampling times were added to Methods: ‘Surveys were conducted between in the spring and early summer (inclusive dates were 28 April to 29 June) during 2017 and 2018’ and the inclusive dates of sampling for each town were added to S1 Table (and referred to in the text). The one-time survey per yard method was clarified: ‘consisted of one, 80-minute timed visual survey of each yard’.

Only until looking at the figure 5, I was not sure if you analysed individual yards or polled them per town. Please, be specific on this in the Methods. I am not sure whether we are not losing too much information by polling yards and using town as observations to be analysed. I understand that this is a simple way how to avoid pseudoreplications, but there are (not so complicated) statistical methods how to solve this. Maybe you should consider to analysed the data at yard scale, which definitely would increase statistical power, but also enable to test some yard specific characteristics as for example yard management (you asked people about it) watering, size etc.

--- To clarify the handling of the data, the following new paragraph was added at the start of the Data analysis section:

‘Data from visual surveys and soil samples were combined; hence abundance and richness include both live snails and shells, unless otherwise specified. In addition, data from the 10 surveyed yards in each town were pooled to produce a single composite data set per town.’

--- With regard to the suggestion of not pooling the yard data, the purpose of this manuscript was to examine large-scale patterns and so pooling the data within each town gave a good representative of each town and evened out some of the with-in town variation. I am lucky that my region of the country has N-S and E-W climatic gradients and this manuscript addresses these gradients.

--- A previous paper addressed land snails and associated yard management (61 yards in a single town; Bergey & Figueroa 2016).

2) Live snails vs empty shells.

It is not clear if only live snails were used to the analyses or snail individuals, including empty shells, were polled and used in all analyses. If so, some test that the live/empty ration is not changes along climate would be useful. Empty shell decomposition rate is highly variable based on climate.

--- See the new first paragraph in the Data analysis section (copied above), which clarifies the combined live snail/shells data.

Getting the actual live snail versus shell ratio would be problematic because of the large number of microsnails, which are hard to categorize as live or dead, especially when samples were viewed well after collection – and numbers of alive-when-collected snails in the soil and shell samples were not recorded. However, I had previously graphed the distribution of live and released snails/slugs during data analysis and included the graph as the new S2_Fig. The following was added to Results:

‘The distribution of live snail abundance (snails counted and released during surveys) was similar to the pattern of all snails (S2_Fig), with relatively low abundance across the northern and eastern tiers, except for the higher relative abundance of live snails in Lawrence (the NE-most town). Centrally located Norman also had a relatively high abundance of live snails compared to all snails. Both live snails and all snails had the highest abundance in the SW-most town of Altus.’

3) The effect of town size.

I realized that the study towns exhibited quite a broad range of sizes, assuming based on town populations ranging from 6.5 to 122K. This is very likely to have an effect on species richness (just as simple SAR pattern) but also more people means more transport, resulting in higher probability of non-native species introduction. I would like to encourage you to test/explore this in your data and to show whether this is or is not an issue.

--- Unfortunately, I was not able to select towns of the same size; however, I avoided Oklahoma City and Wichita, as these are very large metropolitan areas. I also tested for effects of town size when first analyzing the data as I also considered a possible town-size effect, but since size did not correlate well with either abundance or richness, I did not include this in the original manuscript – but have added the suggested regression of population size and snail richness, as follows:

Added to Data analysis: ‘Effects of town population size on snail richness were also tested using linear regression.’

Added to Results: ‘Taxonomic richness was not correlated with town population size (regression: F1,11 = 0.376; p = 0.553; R2 = 0.04).’

I attribute the lack of an SAR effect to (1) The same area (10 yards) was sampled in each town and (2) whereas snails are easily carried, they move slowly on their own, so if introduced in one spot, it’s unlikely that they would be found across town (large towns may indeed have more introduced species, but the distribution would likely be spotty). Had we varied the number of yards that were surveyed in proportion to each town’s population, a species area relationship might have been found.

There might be two levels – the effect of town size/population and the total area cover by the sampled yards within a town. Maybe an interesting analysis, additional to total number of species per town, could be to correlated compositional dissimilarity among yards of the same town with these two size measures and also climate. Are yards in smaller towns more similar or is there any effect of climate (e.g. less humid towns are more homogeneous)?

Related to this point, you should use more than a single regression method as some explanatory variables are very likely to be intercorrelated. Using a simple multilinear regression model we can test the effect of climate and size (population density) on numbers of species and snail abundance. If the size of a town is added as the first variable, then climate is tested on a residual variation, independent of this possibly confounding factor.

--- As discussed above, town population (and probably therefore town size) was not correlated with snail richness (or snail abundance, which I also tested early on, so reanalysis of the climate effects dataset is not warranted. The possibility of compositional variation across yards in different towns is interesting but is not the question addressed by the manuscript and I prefer to keep it ‘tight’.

--- The ‘total area’ is also mentioned a possible factor. I interpret this to be yard area. Please see Bergey and Figueroa 2016, where this variable was tested and found non-significant. My explanation was that snail habitat area (shrubs with leaf litter, debris piles, ground cover plants, …) was important and that this habitat was not captured well by yard area (which includes grass, driveways, decks, …. Grass, which often makes up most of the yard area, was very snail-poor.

4) Distance-decay and Figure 7.

Please improve your theoretical background about distance-decay relationship. The theory says that it is either power law or exponential decay, depending on spatial scale and the homogeneity of species pool. A linear relationship is against the theory and needs to be explained. A good paper to read on this: Nekola, J. C., & McGill, B. J. (2014). Scale dependency in the functional form of the distance decay relationship. Ecography, 37(4), 309-320.

Looking at the data, it is quite obvious that in two cases (A, B), showing a significant decrease of similarity with distance, either power law or exponential function will have a better fit than a linear model.

Please change y-axes to be similarity, so we can see the decrease with distance.

--- I re-analyzed the distance-decay data using an exponential decay regression. The R2 improved slightly, going from 0.33 (linear regression) to 0.35 (exponential decay regression). A new figure showing the exponential decay regression using similarity instead of dissimilarity was added to the manuscript.

I had also mentioned a distance-decay type of pattern with respect to biotic homogenization versus differentiation. As your comment also applies to this, the two instances (in the abstract and in the discussion) were rephrased from ‘distance-decay type relationship’ to ‘distance-related relationship’.

You also need to use Bonferroni/Holm correction of the significance cut level for multiple testing as there are 6 tests run against distance. The correction will certainly leave out only a marginally significant relationship at the D, having visually no decay of similarity with distance. When insignificant, avoid showing a fitting curve, it does not make sense.

--- I made an adjustment for 2 sets of regressions run on non-independent data (similarity based on the full data set and the subset of non-natives; A-C and D-F, respectively, in the now Fig 8; α = 0.025. I indicted that the line in Fig 8D was not significant – see Figure legend). The subsequent two correlations in each column are partitions of the data in A and D that explain the relative contributions of species replacement and nestedness. The method of Baselga (2010) was followed and no correction was made in his peer-reviewed methods paper – nor in subsequent related papers. The lines were retained in all regressions, as these lines provide an easily understood summary of the data and the p values and (now) the indication of significance are given on each graph.

I am hoping you will find these comments useful and wish you good luck with your research.

--- Yes – I appreciate the comments and the resulting greater clarity of the manuscript.

Reviewer #2: I really enjoyed this paper and I'm sure it fills a void in the literature, for several reasons. One is that land snails (and slugs) and grossly understudied. This is esp true for urban habitats. Urban ecology and urban conservation is a rapidly growing field and we know very little about land snails that inhabit cities. In terms of your study, your methodology and research design are excellent. Again, we know very little about climate and precipitation gradients in land snails and this study is the first to tease apart some important patterns. I was esp intrigued by the role that watering lawns seems to play. The distance-decay and homogenization patterns are less surprising but still need to be documented. Your analysis is also very good. I like that you include abundance (Bray-Curtis) and not just species similarity indices. You also did a good job with the taxonomic ID, often a difficult task with snails. Few people rear juveniles to confirm ID.

The paper is well researched, well written and you know the literature well. 

--- Thank you!

I just have a couple of minor suggestions. For line 119, I am not sure what you mean by "composite" soil sample. I am guessing that you sampled in various places but please expand. 

--- Reworded to: ‘In addition to visual searching, a surface soil sample, measuring approximately 0.5 l, was collected in at least two areas with apparent accumulations of micro-snail shells in each yard and, if shells were not apparent, from habitats where micro-snails or their shells were commonly found (e.g., friable soil at the base of buildings). … Visual surveys and soil collections included both front and back yards.’

Secondly, in my copy the text on some figs (1,2,4,6) is blurred. Perhaps that is just my copy but you might check.

--- I have uploaded new figure files, which passed the PACE review for the journal. This should have solved the issue.

---

## [Decision Letter · Decision Letter 1]

11 Aug 2020

Climate gradients, and patterns of biodiversity and biotic homogenization in urban residential yards

PONE-D-20-15300R1

Dear Dr. Bergey,

We’re pleased to inform you that your manuscript has been judged scientifically suitable for publication and will be formally accepted for publication once it meets all outstanding technical requirements.

Kind regards,

Thor-Seng Liew, Ph.D.

Academic Editor

PLOS ONE

Additional Editor Comments (optional):

Both reviewers and me satisfied with the author efforts in improving the manuscript. This is an important contribution to our understanding of urban ecology. Congratulation.

Reviewers' comments:

Reviewer's Responses to Questions

**Comments to the Author**

1. If the authors have adequately addressed your comments raised in a previous round of review and you feel that this manuscript is now acceptable for publication, you may indicate that here to bypass the “Comments to the Author” section, enter your conflict of interest statement in the “Confidential to Editor” section, and submit your "Accept" recommendation.

Reviewer #1: All comments have been addressed

Reviewer #2: All comments have been addressed

2. Is the manuscript technically sound, and do the data support the conclusions?

Reviewer #1: Yes

Reviewer #2: Yes

3. Has the statistical analysis been performed appropriately and rigorously? 

Reviewer #1: Yes

Reviewer #2: Yes

4. Have the authors made all data underlying the findings in their manuscript fully available?

Reviewer #1: Yes

Reviewer #2: Yes

5. Is the manuscript presented in an intelligible fashion and written in standard English?

Reviewer #1: Yes

Reviewer #2: Yes

6. Review Comments to the Author

Reviewer #1: Dear authors,

I am very positive about the revised version. You did a great job and convincingly responded to all my comments.

Thank you very much for the detailed responses and full consideration of my suggestions. I really like your study and think it will be highly cited.

Best regards,

Michal Horsák

Reviewer #2: All my suggested edits have been addressed. In my opinion, this is a very interesting and significant contribution and is ready for publication.

7. PLOS authors have the option to publish the peer review history of their article (what does this mean?). If published, this will include your full peer review and any attached files.

Reviewer #1: No

Reviewer #2: **Yes: **Michael L. McKinney

---

## [Editor Report · Acceptance letter]

18 Aug 2020

PONE-D-20-15300R1 

Climate gradients, and patterns of biodiversity and biotic homogenization in urban residential yards 

Dear Dr. Bergey:

I'm pleased to inform you that your manuscript has been deemed suitable for publication in PLOS ONE. Congratulations! Your manuscript is now with our production department. 

Kind regards, 

on behalf of

Dr. Thor-Seng Liew 

Academic Editor

PLOS ONE